severe mental illness; developing countries; risk factors; barriers; focus groups

**Corresponding author:**
Gerardo A. Zavala;
Email: g.zavala@york.ac.uk

# Identifying barriers and facilitators for health risk behaviours among people with severe mental illness in Bangladesh and Pakistan: a qualitative study

Badur Un Nisa[1,2] (ID), Imogen Featherstone[3], Gerardo A. Zavala[3], Humaira Bibi[1], Md Badruddin Saify[4], Mahmudul Hasan[4], Faiza Aslam[1], Asad Tamizuddin Nizami[1], Rumana Haque[4], Najma Siddiqi[3,5], Richard I.G. Holt[6,7] and Hannah Maria Jennings[3,5]

[1]Institute of Psychiatry, Rawalpindi Medical University, Rawalpindi, Pakistan; [2]Department of Rehabilitation Sciences, Shifa Tameer-e-Millat University, Islamabad, Pakistan; [3]Department of Health Sciences, University of York, York, UK; [4]ARK Foundation, Dhaka, Bangladesh; [5]Hull York Medical School, University of York, York, UK; [6]Human Development and Health, Faculty of Medicine, University of Southampton, Southampton, UK and [7]Southampton National Institute for Health Research Biomedical Research Centre, University Hospital Southampton NHS Foundation Trust, Southampton, UK

## Abstract

People with severe mental illness (SMI) are at greater risk of obesity, cardiovascular disease and diabetes than the general population, due to a higher prevalence of health risk behaviours. Research is needed to inform tailored interventions to improve the health behaviours (diet, physical activity and sleep) of people with SMI in South Asia as these behaviours are closely linked to obesity. The study aimed to explore the barriers and facilitators to healthy diet, physical activity and good sleep among individuals with SMI. A qualitative design was employed using photovoice, semi-structured interviews and focus group discussions. Participants included 16 people with SMI, 16 caregivers and 17 health professionals in Bangladesh and Pakistan. Data were analysed thematically, informed by the socio-ecological framework. A complex interplay of individual, familial and societal factors influenced these health behaviours. Individual factors include knowledge, beliefs and mental health limitations. Caregivers play a crucial role in influencing behaviour. At the societal level, gender expectations, financial constraints and religious influences significantly impact these behaviours. The insights from this research can inform tailored interventions for this vulnerable group and highlight the need for integrated services, financial support and improved urban planning.

## Impact statement

This research offers critical insights into addressing the growing obesity and physical health risks among people with severe mental illness (SMI) in South Asia by identifying a range of factors that influence the health behaviours (healthy eating, physical activity and sleep) of people diagnosed with SMI in Pakistan and Bangladesh. By identifying key individual, sociocultural and systemic barriers to healthy behaviours, our findings provide a foundation for designing culturally appropriate and gender-sensitive interventions. These interventions have the potential to improve not only the physical health outcomes of individuals with SMI but also the overall quality of life. The involvement of family caregivers is crucial. At a policy level, integrated health services, financial support for healthy living and improved urban planning are needed to create safe spaces for physical activity. The impact of this research extends beyond clinical practice to influencing health policy and urban planning, offering a holistic strategy to reduce health disparities among people with SMI in the region.

## Introduction

Severe mental illness (SMI) refers to a group of conditions, including schizophrenia, bipolar disorder, schizoaffective disorder and major depressive disorders with psychotic features, all of which decrease the ability of a person to perform daily life activities (Public Health England, 2018; Carolan et al., 2019). People with SMI are 2.5 times more likely to have obesity than the general population, with the global prevalence of obesity among individuals with SMI estimated to be 25.9% (Afzal et al., 2021). This heightened risk can be attributed to factors including medication

side effects, poor dietary choices, reduced physical activity and poor sleep (McKibbin et al., 2014; Barnard-Kelly et al., 2022). Obesity plays a significant role in the development of non-communicable diseases (NCDs), such as cardiovascular disease and diabetes, which are leading causes of mortality among people with SMI ('World Bank Open Data', n.d.; Bahorik et al., 2017; Mueller-Stierlin et al., 2022).

The prevalence of obesity is higher in high-income countries, but, more recently, there has been a steep rise in people with SMI in low- and middle-income countries (LMICs), particularly in the South Asian region (Afzal et al., 2021; Zavala et al., 2022). According to World Health Organization (WHO) international BMI cut-offs, 16% of people with SMI in Bangladesh, India and Pakistan are obese, and 30.2% are overweight; according to WHO Asian BMI cut-offs, these figures rise to 46.2% obese and 17.3% overweight, respectively (Appuhamy et al., 2023).

Improvement in diet and physical activity is part of the WHO 'Best Buys' strategies to prevent NCDs (Allen et al., 2018; Zavala et al., 2022). Sleep plays a crucial role in weight regulation and metabolic health, with poor sleep quality and duration linked to increased risk of obesity through mechanisms such as hormonal imbalances, increased appetite and reduced energy expenditure (Miller et al., 2021).

People with SMI experience significant challenges in engaging with behavioural interventions compared with the general population (Scott and Happell, 2011). It is important to understand the specific barriers faced by this population in adopting and sustaining healthier behaviour, including a healthy diet, regular physical activity and good sleep, to tailor interventions to their needs and preferences (Balogun-Katung et al., 2021). Identified barriers to healthy behaviour from high-income countries include limited health knowledge, low motivation, restricted access to safe spaces for physical activity as well as financial and social barriers (Deenik et al., 2019; Teychenne et al., 2020; Balogun-Katung et al., 2021; Hassan et al., 2022). However, there is a lack of research exploring the barriers and facilitators to improving healthy behaviour among people with SMI in LMICs, particularly in South Asia. Additionally, exploring the perspectives of relevant stakeholders, such as people with SMI, their family caregivers and healthcare providers, is essential to inform the development of interventions that are better aligned with their needs.

The aim of this study was to explore the barriers and facilitators to healthy diet, physical activity and good sleep among individuals with SMI at individual, community and systems levels. We sought to explore current practices and develop an understanding of the perspectives of people with SMI, their caregivers and healthcare providers on the challenges and supportive factors to improve these health behaviours.

## Methods

We report this study according to the Standards for Reporting Qualitative Research (Appendix 1 in O'Brien et al., 2014).

### Research design

This qualitative study utilised the socio-ecological framework to explore influences on healthy behaviours (related to diet, physical activity and sleep), and intervention development. The framework recognises the complex interplay between individual and environmental influences on behaviours, and using it enabled a structured analysis of influences on healthy behaviours at the individual, relationship, physical and sociocultural environmental levels (McLeroy et al., 1988).

For data collection, we utilised photovoice with in-depth interviews (IDIs) and focus group discussions (FGDs) (see Supplementary material 1 and 2). Photovoice is a community-based participatory research method where participants can identify, represent and convey their perspectives using photographs (Budig et al., 2018). It has been used with a range of groups, including people with impaired cognitive functioning and mental health problems (Povee et al., 2014).

### Setting

Data were collected at the National Institute of Mental Health and Hospital in Dhaka, Bangladesh, and the Institute of Psychiatry Benazir Bhutto Hospital in Rawalpindi, Pakistan. Both are tertiary care government institutions providing essential mental health services to a broad range of populations, including urban and rural communities, in their respective regions. There are currently no dedicated nutrition or healthy lifestyle-related services for individuals with SMI at either of these hospitals. There was a nutritionist at the site in Pakistan, employed for a specific time-limited research project, who provided nutritional support to people with SMI. These settings were selected to reflect diverse yet comparable South Asian contexts. Using two sites allowed exploration of both shared and setting-specific influences on health behaviours among people with SMI. Standardised methods were applied across sites, including shared topic guides, researcher training and joint analysis to ensure consistency while acknowledging contextual nuances.

### Participants and recruitment

We purposively recruited three groups of participants: (1) patients with SMI, (2) caregivers and (3) health professionals. Patient participants were outpatients (≥18 years), diagnosed with SMI according to the *Diagnostic and Statistical Manual of Mental Disorders, Fifth Edition*, by their treating mental health practitioner, and had capacity to provide informed consent. We sought to include men and women with a range of SMI diagnoses. Caregivers were family members residing with the patient for at least 6 months. Healthcare providers had at least 6 months experience in providing mental health care. We sought to include a range of health professional roles. The sample size was based on the approximate number required to achieve data saturation, as observed in previous studies focusing on people with SMI (McKibbin et al., 2014).

#### Participant characteristics

There were 49 participants, 25 from Pakistan and 24 from Bangladesh, including 16 participants with SMI, 16 caregivers and 17 health professionals (Table 1).

Participants with SMI were primarily selected from a database of individuals who had previously consented to be re-contacted for research (Rajan et al., 2023). Additional patients and their caregivers were recruited from Psychiatry Outpatient clinics. The capacity to provide informed consent was assessed verbally by trained research staff using a checklist to confirm participants' understanding of the study, including its purpose, procedures, their right to withdraw and the use of audio recording. This process was guided by principles similar to those outlined in the Mental Capacity Act, ensuring that participants could understand, retain, use and communicate information relevant to their decision to participate. The

**Table 1.** Participant characteristics

| Total participants; *n* = 49 (25 from Pakistan and 24 from Bangladesh) | | | |
|---|---|---|---|
| People with SMI (*n* = 16) | Age (years) | Range | 25–62 |
| | | Median | 41 |
| | | IQR | 14.5 |
| | Sex (male/female) | *M* = 11, *F* = 6 | |
| | Diagnosis | Major depressive disorder | 4 |
| | | Psychosis | 5 |
| | | Bipolar | 6 |
| | | Schizophrenia | 1 |
| | BMI | Range | 19–33.7 |
| | | Median | 25.8 |
| | | IQR | 6 |
| Caregivers (*n* = 16) | Age | Range | 23–63 |
| | | Median | 42.5 |
| | | IQR | 27.5 |
| | Sex (male/female) | *M* = 6, *F* = 10 | |
| | Relationship with the participants with SMI | Parent | 4 |
| | | Spouse | 4 |
| | | Son/daughter | 2 |
| | | Sibling | 3 |
| | | Sister | 1 |
| | | Uncle | 1 |
| | | Daughter-in-law | 1 |
| Healthcare providers (*n* = 17) | Sex (male/female) | *M* = 9, *F* = 8 | |
| | Age | Range | 24–53 |
| | | Median | 37.5 |
| | | IQR | 21 |
| | Participant role | Psychiatrist | 6 |
| | | Psychologist | 5 |
| | | Staff nurse of psychiatry | 2 |
| | | Nutritionist | 2 |
| | | Intern | 1 |
| | | Psychiatric social worker | 1 |

*Abbreviations: BMI, body mass index; IQR, interquartile range; SMI, severe mental illness.*

Head of Department suggested that healthcare providers be invited to participate based on the inclusion criteria. All potential participants were provided with a study information sheet. The study was explained to them, and they had the opportunity to ask questions before written informed consent was taken.

### Data collection

Data collection methods included photovoice, interviews and FGDs:

#### Photovoice

Participants with SMI and their caregivers participated in a 2 hour workshop. The research and the photovoice approach were explained, and participants could practise taking photographs.

Participants with SMI were asked to take photographs with a mobile phone every day for 7–14 days. They were asked to take photographs, with caregiver support if needed, of facilitators and barriers to their physical activity, healthy diet and sleep behaviours. They were requested not to include photographs of people's faces. Participants brought their phones to the interviews, where anonymised images were shared with the researcher *via* WhatsApp or a USB cable. The interviews were then conducted, during which the photos were discussed (see below). All anonymised photos were securely transferred and stored on a password-protected private server.

#### Interviews and focus group discussions

Researchers in Pakistan (BUN and HB) and Bangladesh (MBS and MH) conducted the interviews and FGDs (November 2023 to February 2024). Topic guides were developed based on the research objectives and the socio-ecological framework. The interview topic guide was piloted with one participant.

Participants with SMI took part in face-to-face semi-structured interviews (25–90 minutes duration). The topic guide included prompts for discussing their photographs and questions about their current diet, physical activity and sleep and the barriers and facilitators to healthy behaviours.

Two FGDs (2 hour duration) were conducted at each research site, with caregivers and health professionals. In addition to the topics explored with participants with SMI, the caregivers' and health professionals' roles in supporting healthy behaviours among people with SMI were explored. The FGD for health professionals also discussed current relevant policies and programmes, as well as strategies for future intervention development.

### Data management and analysis

FGDs and individual interviews were digitally recorded, translated from Urdu and Bangla to English and transcribed verbatim. Transcription and translation were conducted simultaneously by bilingual research assistants fluent in both the local language (Urdu or Bangla) and English. An independent bilingual researcher reviewed all transcripts for accuracy and quality. Additionally, selected excerpts from two transcripts were back-translated into the original language to assess the fidelity of the translation process and ensure the preservation of culturally nuanced meanings. The photographs and transcripts were anonymised, and data were shared securely on a password-protected private server with the research team.

Thematic analysis was used to identify, analyse and interpret themes from the qualitative interview and focus group data (Braun and Clarke, 2006). NVivo 20 software supported the analysis ('NVivo Leading Qualitative Data Analysis Software (QDAS)', 2022). We used elements of both 'data-driven' and 'theory-driven' approaches (Fereday and Muir-Cochrane, 2006). Two researchers (IF and HMJ) carried out independent line-by-line open coding of the same interviews before comparing coding and inductively developing an initial coding framework. This was structured in relation to behaviours (healthy diet, physical activity and sleep), the influences on these behaviours, current service provision and intervention development. Four researchers (IF, HMJ, BUN and HB) coded the remaining transcripts using this framework, which was iteratively developed during this process: using memos to record

coding queries and changes, which were regularly reviewed and agreed through team discussions (HMJ, IF, BUN and HB). IF then used the socio-ecological framework to organise the codes regarding influences on healthy behaviours and intervention development. The analysis team (IF, HMJ, BUN and HB) subsequently reviewed and summarised the content of the codes, and the relationships between them, to develop descriptive themes and subthemes. The analytic themes were then written up and reviewed by the analysis team. Trustworthiness and credibility of the analysis were enhanced by the use of 'constant comparison' and the involvement of several researchers (Lincoln and Guba, 1985).

### Reflexivity

The research team included members with relevant expertise in mental health and physical comorbidities research, context-specific knowledge of health service provision, language and culture in South Asian contexts and qualitative research methods. This enabled the research to be conducted rigorously and enhanced its findings.

### Results

The findings are organised according to eight themes, which report on current practices regarding diet, physical activity and sleep, as well as the individual, relationship and societal-level influences on these behaviours among participants with SMI.

### Current behaviours and support regarding healthy eating, physical activity and sleep

Participants with SMI described eating traditional foods: rice or *roti* (flatbread) with vegetables, lentils and various curries. Spicy and fried foods were popular. Breakfasts often included *paratha* (fried flatbread). Snacks included biscuits, sweets or fried snacks with tea or fizzy drinks.

IDIPK06 (Person with SMI): "I eat biscuits, candies… and like chips… I also consume juice and fizzy drinks."

Some participants ate healthy foods (vegetables, fruit and nuts) when able. Some had regular meal times, while others ate whenever they felt hungry.

Participants with SMI mostly engaged in physical activity during routine daily tasks. For men, this commonly included paid work or walking as a means of transport, and for women household chores. Some men described playing sports occasionally.

Many participants with SMI reported going to bed between 9 and 11 pm, although phone usage and work commitments often delayed this. They commonly woke for early morning prayers and some took daytime naps. Many described experiencing significant sleep disruption due to mental health symptoms and sleep environment.

Participants with SMI described receiving advice from health professionals, family and social media. Psychiatrists in both countries said that they commonly gave brief advice on healthy eating and physical activity during their consultations.

PKHCP4 (Psychiatrist): "It is difficult for me to give a long session to them, so I simply suggest trying to avoid sugar-coated and fried things and eating more fruits and vegetables in your daily routine."

Psychiatrists and psychologists in Pakistan described providing a range of strategies, including scheduling activities and mealtimes and taking a graded approach to increasing physical activity. Psychologists in Pakistan provided Cognitive Behavioural Therapy and Behavioural Activation interventions that include work on healthy behaviours.

Psychiatrists in both Pakistan and Bangladesh were well informed regarding sleep hygiene strategies and reported routinely advising patients and their families, and providing sleep medication.

### Knowledge and beliefs regarding healthy behaviours (individual level)

Most people with SMI knew fruit and vegetables are nutritious choices, and processed and high-sugar foods are less prudent choices. However, views varied on the nutritional value of other foods, including meat, rice and bread. Some understood that oily, fried foods were less prudent choices, while others did not. Overall, a lack of specific dietary knowledge was identified by both people with SMI and healthcare providers.

PKHCP9 (Psychologist): "We don't give any specific healthy diet plan to patients. We don't have any specific knowledge related to diet and nutrition of food."

Cultural beliefs influenced eating, for example, some perceived a fat tummy as a sign of health. A widely held belief among people with SMI was that home-made food is healthy. This was partly due to associating it with simple, fresh ingredients. However, some did not recognise that home-made foods, such as fried foods, could also be less prudent choices. They commonly interpreted 'healthy food' as food that was hygienically prepared and did not cause stomach problems. This strongly contributed to the belief that home-made food was healthier than food prepared 'outside'.

IDIPK01 (Person with SMI): "It is a healthy food, it is a good homemade food…we are tired of outside food as it can cause stomach problems."

People with SMI identified many benefits of healthy eating, including improving physical health, energy levels, mental alertness and physical appearance, and a range of health consequences of unhealthy diets.

Some participants with SMI reported being motivated in their food choices and to reduce portions by the desire to 'eat healthy'. However, knowledge did not always motivate them to eat healthily. Taste and enjoyment of food were also powerful motivators, and both caregivers and people with SMI said that they found it difficult to be disciplined.

IDIBD04 (Person with SMI): "Unhealthy food tastes good. That's why people eat it knowing that it is harmful."

Many people with SMI identified physical and psychological benefits of physical activity including distraction from disturbing or anxious thoughts, reducing stress, feeling more relaxed and positive and being fit to work. Consequences of being inactive included becoming unhealthy, gaining weight, pain and tiredness.

IDIBD03 (Person with SMI): "When you play sports, both the body and mind stay well. Tension can be relieved."

Most participants with SMI considered sleeping continuously from approximately 10/11 pm until morning a good pattern. The benefits of good sleep identified by participants included feeling fresh, the ability to think clearly, improved mood, good physical health and the ability to work and engage in the next day's activities. Reported consequences of poor sleep included feelings of irritability, anger, anxiety, difficulty speaking clearly, feeling physically unwell, difficulty concentrating and a negative impact on the next day's activities.

IDIPK04 (Person with SMI): "There are many benefits to a good night's sleep. Our minds will be fresh, our health will be good, our physical skin will be affected, and our mood will be good."

People with SMI and caregivers described a range of strategies to enable better sleep, including a regular sleep routine, not staying up late or sleeping during the day, work or activity during the day, a peaceful sleep environment, comfortable clothes, not using phones near bedtime, praying, drinking milk and not eating near bedtime and medication.

Those with SMI said that it would be helpful to learn more about healthy behaviours. Some people with SMI and caregivers suggested that doctors or experts (nutritionists and research organisations) could provide guidance. For intervention delivery, they suggested using videos, booklets and in-person or online group sessions.

### Mental health influences on healthy behaviour (individual level)

All participant groups described how mental health issues and medications influence diet and physical activity. Low mood and medications reduced the appetite of some participants, while others experienced increased appetite and over-eating.

IDIPK04 (Person with SMI): "Stress affects me a lot. If I have any tension I can't eat."

Low mood decreased the energy and motivation of those with depression to engage in physical activity, while activity levels of individuals with manic or psychotic symptoms varied. Having psychotic symptoms and anxiety could be a barrier to being physically active outside.

IDIPK09 (Person with SMI): "(I) locked myself in a room for almost 10 months because I was very scared and anxious and avoided the people and things outside of my room."

People with SMI reported that their mental health affected their sleep, and the quality of their sleep affected their mental health. Symptoms disrupting sleep included anxiety, fear and mental disturbance.

IDIPK06 (Person with SMI): "I used to stay awake until 3 or 4 am. I felt terrified and anxious."

In developing an intervention, health professionals identified that some people with SMI have cognitive difficulties or limited capacity to understand and follow advice. Therefore, any intervention needs to be adapted accordingly.

PKHCP2 (Psychiatrist): "It depends on how much patient understanding you have then you can tell the patients how carbohydrates and fats work... how I tell these things according to patients' understanding level is very important."

Difficulties in communicating and interpersonal relationships due to mental illness were described by those with SMI and observed during interviews. For example, when asked what she ate on a typical day, a participant (IDI_PK_02) experiencing psychosis responded, "I eat the truth," and said that the interviewer was a liar and a thief. Health professionals explained that people with manic or psychotic symptoms may lack 'insight', affecting their ability to follow instructions and their motivation to engage in interventions. Giving time and attention to people with SMI and building a rapport with them was identified as important for successful intervention delivery.

### Influence of physical illness on healthy behaviour (individual level)

People with SMI described ways in which physical illness influenced their behaviour. Some avoided certain foods, including healthy foods, due to physical health conditions such as digestive and dental problems. Some with conditions such as diabetes or cardiovascular disease chose to eat more healthily as a result, but others found it difficult to control their diet.

IDIPK07 (SMI participant): "Being human sometimes it's not easy to control yourself like I am diabetic and also fond of sweets."

Many people with SMI said that their physical activity was limited due to issues such as pain, breathing issues, constipation and urinary incontinence, heart problems and diabetes complications. A few described sleep disturbance due to pain and continence issues.

### Influence of family on healthy behaviours (relationship level)

People with SMI commonly ate meals prepared for the whole family; therefore, the preferences of others influenced what they ate and how healthy it was. As the women in the family, including those with SMI, usually cooked meals, male participants had less control over what they ate. It was only in exceptional circumstances that their meals were not prepared by female relatives. For example, a participant (IDIPK09) whose mother had died explained that his father did limited cooking and they therefore ate from restaurants and hotels despite his belief that this food made him ill. Although some people with SMI went food shopping, this was more commonly done by their family caregivers, so the latter would make food choices.

Some people with SMI had night-time caring responsibilities for children or relatives. Many slept in the same room as family members and described this as disruptive to their sleep.

IDIPK01 (Person with SMI): "One of my siblings uses the phone while the other sits talking to each other, and there is only one room in which all four to five people sleep."

The high cultural value placed upon family was highlighted by health professionals as supportive in caring for people with SMI.

BGHCP4 (Nutritionist): "It is our strength that in our society, family bonds are still intact and we still value our families... we can return the patient to their family and the family can take care of the patient."

Families were reported to be important in supporting healthy behaviours. Many people with SMI described their motivation to engage in physical activity increased when with family or friends. Health professionals explained that family caregivers can help overcome the barriers to physical activity related to mental health conditions, including low motivation, difficulty following instructions and anxiety about going outside. Health professionals reported advising family caregivers to support patients in good sleep and eating habits. Some people with SMI said that their families' advice influenced their choices, while others did not follow their advice.

IDIBD06 (Person with SMI): "I don't listen to anyone at home. This is my problem."

However, some people with SMI did not have a family to support them or they were too busy.

Health professionals identified that increasing family caregivers' understanding of mental illness and healthy behaviours was vital to enable them to provide support.

### Cultural norms and beliefs (societal level)

Gender norms, stigma, concerns about safety and religious beliefs all influenced behaviour.

Cultural gender norms regarding meal preparation influenced food choices for people with SMI (as described above), as well as the physical activity that men and women engaged in. Many men with SMI explained that their main physical activity was work outside

the home, ranging from strenuous manual labour to walking around the office. They commonly walked to work or the market, and some walked for exercise or pleasure. The main physical activity of most women with SMI was during household chores.

PKCG5 (Caregiver): "My mother's physical activity is all about doing household tasks like she regularly does dishwashing, cleaning, cooking, laundry, *etc.* which demands a lot of physical movement."

Gender barriers to women being physically active outside included the potential for sexual harassment. Both women and people with SMI experienced stigma and safety concerns.

IDIBGCG2 (Family carer): "There is no way to go for a walk for girls because of eve teasing" (the making of unwanted sexual remarks or advances by a man to a woman in a public place).

PKPA3 (Health professional): "If I told patients to walk outside the home then they may face stigmatisation."

Religious rituals were a significant source of physical activity for many with SMI, for example, going to the mosque to pray and actively participating in religious festivals. Many experienced improved sleep quality after prayer.

IDIPK04 (Person with SMI): "When I pray, I get peace of mind. I sleep peacefully. And until I pray I don't get sleep."

However, too much involvement in religious activities, such as excessive recitation of verses, could disrupt sleep. Health professionals suggested that maintaining a balance in these practices was important.

### Financial influences (societal level)

All participant groups highlighted the significant impact of financial constraints on healthy eating. Affordability issues led to compromises in food quality, causing people to prioritise cheaper, less nutritious options. Many described the high cost of fresh and dry fruits as prohibitive, or prioritised their children eating them. Some said that the rising cost of essential ingredients, such as vegetables and meat, made them unaffordable.

BGCG2 (Caregiver): "Sometimes we cannot buy vegetables because of lack of money, let alone meat."

People with SMI and their family caregivers cited financial constraints as a barrier to engaging in structured exercise. However, saving money motivated some participants to walk, instead of using transport.

Health professionals faced challenges in recommending healthy foods due to their high cost. They emphasised the importance of addressing financial difficulties to promote healthy behaviours, suggesting interventions such as financial support for healthy food and advocating for government initiatives to address food insecurity and promote physical activity among socio-economically disadvantaged individuals.

### Government, policy and service-level challenges and solutions (societal level)

Areas identified that need policy-level change to facilitate healthy behaviour included more regulations on food adulteration, particularly in Bangladesh. The need for city planning to reduce traffic and pollution and to provide more clean, safe spaces and parks for exercise (particularly for women) was highlighted, as was the need to provide more monetary and social support for patients with SMI.

A number of challenges to people with SMI accessing services were raised by health professionals, including a lack of services, costs and stigma. They reported that doctors had limited time to

spend with their patients (seeing up to 60 patients a day). In both countries, mental health services were reported not to be integrated with physical health services. There were no policies or guidelines around exercise or diet for people with SMI, although there were guidelines related to sleep. It was suggested that having such guidance would help healthcare professionals integrate appropriate advice into their practice.

PKHCP1 (Psychiatrist): "Yes there are guidelines to see diabetic patients, but for SMI patients there are no policies related to diet."

In both countries, the need for better integration between different health professionals and services was recognised.

PKHCP9 (Psychologist): "There should be a team that has nutritionists and a social worker. There should be a more integrated approach."

While many barriers were identified, in Bangladesh, health professionals believed that the recent mental health policy and plan introduced in 2024 offered real opportunities for change, including more training and development of services and interventions for people with SMI.

BGHCP3 (Psychiatric social worker): "I think our main advantage is acknowledgement from the state…we do not face any difficulties from the authorities, rather they make us work properly."

## Discussion

### Key findings

This study identified barriers and facilitators to maintaining a healthy diet, engaging in physical activity, and achieving good sleep patterns among people with SMI in Bangladesh and Pakistan and to developing interventions for this population. On the individual level, significant barriers included mental health symptoms and medication side effects that reduced energy and motivation, while facilitators involved knowledge and beliefs about the benefits of healthy behaviours. Family caregivers were important facilitators in supporting these behaviours, providing motivation and practical support. Environmental barriers were also critical, such as noisy, shared sleeping conditions and a lack of facilities for physical activity. At the societal level, gender expectations and financial constraints significantly impacted these health behaviours, with religious influences playing both supportive and non-supportive roles. Furthermore, there was a notable lack of services or interventions designed to support individuals in making these behavioural changes. The absence of adequate support services exacerbates these challenges, leaving individuals with SMI without the necessary resources to improve their health outcomes effectively.

### Explanation of the findings

We found knowledge and beliefs about healthy behaviours to be inadequate, with both people with SMI and health professionals showing limited understanding of healthy eating and nutrition. Mental health symptoms, such as low energy and motivation, were frequently cited as significant barriers to engaging in healthy behaviours. These findings align with other studies in South Asia, which show that people with SMI are less likely to adopt healthy behaviours due to a complex interaction of cognitive impairments, low motivation and the side effects of psychotropic medications (Thongsai et al., 2016; Appuhamy et al., 2023; Zavala et al., 2023b). Moreover, these symptoms often exacerbate unhealthy behaviours,

such as poor diet and sedentary lifestyles, contributing to poor physical health outcomes, such as obesity and cardiovascular diseases (Appuhamy et al., 2023). Compared with general populations in South Asia, individuals with SMI face greater challenges due to their compounded mental health issues and the socio-economic barriers, including financial limitations, which were highlighted in this study (Teasdale et al., 2017). The lack of access to health-promoting resources, combined with cultural beliefs and inadequate social support systems, creates a cycle of poor mental and physical health (Zavala et al., 2023b).

In contrast, some studies conducted in higher-income countries show that while people with SMI also struggle with engaging in healthy behaviours, they often have better access to structured interventions aimed at addressing these challenges (Zavala et al., 2022). This disparity in access to health services and interventions further emphasises the importance of context-specific interventions that address not only mental health symptoms but also the socio-economic and cultural factors that shape health behaviours in South Asia (Zavala et al., 2023a).

Additionally, while physical activity was mentioned by many participants, it was primarily described in the context of unstructured daily activities, such as walking, household chores, and labour-intensive work. These forms of activity were incidental and not planned or structured, with no participants reporting regular or formal exercise routines. For instance, women referred to housework as their primary form of activity, whereas men spoke of walking or job-related tasks, with only a few mentioning occasional sports. These descriptions lacked information on frequency, intensity or duration, suggesting that physical activity was shaped by environmental and socioeconomic constraints rather than by deliberate health choices.

The crucial role of family caregivers was anticipated, given the strong cultural emphasis on family in South Asia, as seen in similar studies by our and other research groups (Zavala et al., 2023a; Turiho et al., 2024). However, the severe impact of financial constraints on dietary choices and physical activity was more pronounced than expected. Compared with the general population in Pakistan and Bangladesh, individuals with SMI face amplified challenges due to their mental health conditions (Khan et al., 2022; Appuhamy et al., 2023). Financial constraints, while common in the broader population (Dizon, 2021), are exacerbated in individuals with SMI by factors such as medication side effects and the impact of mental health symptoms on their motivation, energy levels and ability to work (Ashdown-Franks et al., 2018). These findings show the intersection of socio-economic and health-related barriers, illustrating the compounded difficulties faced by this vulnerable population. Additionally, the lack of supportive services further compounds these difficulties, indicating a significant gap in the healthcare system that needs to be addressed.

### Implications

There is a need for integrated health interventions that address both mental and physical health in people with SMI. Policies should aim to alleviate financial barriers by providing financial planning, subsidies or financial support for healthy foods and physical activity programmes. Moreover, city planning must prioritise the creation of safe, accessible spaces for physical activity, and interventions should consider the additional cultural challenges experienced by women to engage in physical activity. Policymakers should leverage the supportive aspects of religious practices while mitigating any negative impacts on sleep and daily routines.

Health services should be equipped to offer health risk modification interventions tailored to the cultural context, involving family caregivers in the process. Education on healthy diet and physical activity, as well as adequate time and resources, is needed to enable healthcare providers to engage people with SMI and their family carers in these interventions. The delivery of interventions must be adapted to the needs of people with SMI, which may include reduced motivation, cognitive and interpersonal difficulties.

### Strengths and limitations

A strength of this study is its use of the photovoice method, which empowered participants to document and share their experiences, providing rich, user-centric insights into their daily lives. This approach facilitated a deeper understanding of the context-specific barriers and facilitators to healthy behaviours. Our use of the socio-ecological approach enabled us to undertake structured analysis of the influences on health behaviours on multiple levels and to explore the complex interplay between individual and sociocultural influences. Additionally, the study's focus on both Bangladesh and Pakistan allowed for a comparative analysis within the South Asian context and the possibility of evaluating commonalities that could be used for intervention development in both settings. However, the study had several limitations, including a low proportion of participants with obesity, which may affect the generalisability of findings related to obesity. The qualitative design, while providing in-depth insights, may not be representative of all individuals with SMI in these countries. The translation and analysis of data into English, while necessary for practical reasons, was a weakness of the study. When analysing translated data, meaning – particularly nuances – can be missed. However, as outlined in the 'Data management and analysis' section, we took several steps to mitigate this risk.

Due to the relatively small sample size (appropriate for a qualitative study) and resource limitations, it was not possible to explore in depth all potential individual and contextual characteristics that may have influenced the experiences of participants with SMI and their health behaviours, such as age, duration of illness and education. The findings did, however, reveal important gender differences and the impact of comorbidities on health behaviours. Further qualitative research that includes more diverse samples is warranted. In addition, quantitative studies could complement these findings to provide a more comprehensive understanding of the issues faced by individuals with SMI.

### Conclusion

We found a complex interplay of individual, familial and societal factors influencing health behaviours among people with SMI in Bangladesh and Pakistan. There is a need to develop interventions that increase the understanding of people with SMI and their family caregivers of healthy behaviours and that are tailored to their mental health needs. There is a critical role for family support in intervention delivery. There is a substantial impact of financial and environmental barriers on changing health risk behaviours. There is a need for integrated health services, financial support for healthy living and improved urban planning to create safe spaces for physical activity. Overall, this research provides insights into factors that should be considered for the development of effective, context-

specific interventions aimed at improving the health and well-being of people with SMI in South Asia.

**Open peer review.** To view the open peer review materials for this article, please visit http://doi.org/10.1017/gmh.2025.10016.

**Supplementary material.** The supplementary material for this article can be found at http://doi.org/10.1017/gmh.2025.10016.

**Data availability statement.** Raw data are contained within this manuscript in the form of participant quotes and field note extracts. The corresponding author is available to contact for further information. However, further data about each specific data extract cannot be provided due to the risk of re-identifying the participants involved.

**Acknowledgements.** We would like to express our gratitude to all study participants for their time and commitment. We are also grateful to the doctors, administration, and support staff at the Institute of Psychiatry (IoP), Rawalpindi, Pakistan and the National Institute of Mental Health and Hospital (NIMH), Dhaka, Bangladesh for their continuous support in conducting the study.

**Author contribution.** GAZ, HMJ, MH, FA, ATN, RH and NS contributed substantially to the conception or design of the work. BUN, HB, MBS and MH were responsible for data acquisition. IF, HMJ, BUN, HB and RIGH contributed to data analysis and interpretation. The manuscript was drafted by IF, BUN, HMJ, MBS, GAZ, HB and RIGH. All authors critically revised the manuscript for important intellectual content and approved the final version to be published. BUN and IF are the joint first authors.

**Financial support.** The funders had no role in the design of the study; in the collection, analysis or interpretation of data; in the writing of the manuscript; or in the decision to publish the results. This study was supported by an internal grant from the University of York (Grant No. 4132949).

**Competing interests.** The authors declare no Competing interest.

**Ethical standard.** This research adhered to strict ethical principles and guidelines throughout its conduct. Ethics approval was granted by the University of York (HSRGC/2023/576/E), Rawalpindi Medical University (520/IREF/RMU/2023), Institute of Psychiatry (IOP) and the ARK Foundation of Bangladesh (ARK/Research/2O23/0089). Informed consent was obtained from all participants, ensuring they were fully aware of the research's objectives and procedures. Voluntary participation was emphasised, and participants had the freedom to withdraw from the study at any point without repercussions. To minimise harm, all necessary precautions were taken to ensure participants' comfort and safety during the research process. Confidentiality and anonymity were maintained to safeguard the privacy of participants, with any personally identifiable information removed from transcripts.

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
