## [Reviewer Report]

This qualitative study in two psychiatric state hospitals in Bangladesh and Pakistan examines barriers and facilitators to health behaviors amongst persons with severe mental illness (SMI). Photo-voice supported interviews with persons with SMI and focus group discussions with caregivers and mental health providers are used, along with a socioecological framework to interpret results and make recommendations at institution and policy levels. Given the scarcity of data from LMIC and South Asian context on the topic, this is a useful and important piece of research.

Some minor clarifications and changes are recommended. Could there be light shed on current systemic support or infrastructure or health behaviours for the persons with SMI in these two countries/hospitals? I would have assumed there to be no such services, and that is what is highlighted in the discussion, but I do see nutritionist listed as a healthcare provider interviewed, which appears contradictory to these statements. This information on current provisions could be shared in the introduction, along with setting. Thus when suggesting interventions and policy recommendations in the discussion, the gap that needs to be filled can be better understood.

Minor language clarifications -

Suggest participants numbers currently in “results” be moved to methods as numbers reference would be helpful when justifying sample size (line 79-88).

Line 18-20: People with SMI are 2.5 times more likely to have obesity 19 than the general population, with the global prevalence estimated to be 25.9% (Afzal et al. 20 2021). -> please edit the sentence, global prevalence estimate of what is 25.9%? It is unclear

suppression is chained in a free country - the quote by person with SMI, line 331- clarify as currently sentence not comprehensible

---

## [Reviewer Report]

1- Rationale for Multiple Study Sites

The rationale for including two study sites is unclear. Given the likely contextual and demographic differences between the two settings, one would expect significant variability in participant experiences and responses. It would be important to justify this decision and explain how potential inter-site differences were accounted for in the study design and analysis.

2- Data Collection Sequence and Translation Concerns

Could you clarify the sequence of data processing—does it follow the order: recording → translation → verbatim transcription? If so, there may be implications for the fidelity of the data. Analyzing translated texts can result in loss or distortion of meaning, particularly for culturally nuanced language. Please elaborate on how this issue was addressed or mitigated during analysis.

3- Definition and Details of Physical Activity

The description of physical activity from the perspective of participants with severe mental illness (SMI) would benefit from greater detail. For example, how frequently did participants engage in physical activity (e.g., number of days per week, duration)? Was the activity structured or informal? Providing this context would enhance the interpretation of the findings.

4- Participant Capacity and Informed Consent

One participant reportedly expressed paranoia toward the interviewer and made the statement that she had “eaten the truth.” This raises important ethical concerns regarding her capacity to provide informed consent. Was her mental state assessed to ensure she was stable enough to participate in the study? A brief explanation of how capacity to consent was evaluated would be helpful.

5- Inclusion of Non-Communicable Diseases (NCDs)

Did the sample include individuals diagnosed with non-communicable diseases (NCDs)? If so, were any specific insights or patterns observed in this subgroup? Clarifying this aspect could add further depth to the findings.

6- Consideration of Personal and Contextual Factors

The interpretation of findings could be strengthened by exploring associations with relevant personal factors, such as age, duration of illness, changes in pre-morbid functioning, level of education, and income. Including such variables would help contextualize the results and identify any meaningful sub-group differences.

---

## [Editor Report]

Dear authors,

Below you will find the reviewers' comments. Overall, the reviewers found your paper compelling and informative. However, additional clarifications are required, as noted below. Please revise the manuscript thoroughly and prepare a point-by-point response letter.

I look forward to receiving a revised version.

---

## [Reviewer Report]

Thanks, this looks good. Please fix punctuation end of line 354 - change comma into colon or otherwise alter grammatical structure as preferred.